# QUESTIONING SIMPLICITY BIAS ASSUMPTIONS

## ABSTRACT

The Simplicity Bias (SB) is the observation that the training of most commonly used neural network architectures with standard training techniques is biased toward learning "simple" functions. This phenomenon can be a benefit or drawback depending on the relative complexity of the desired function to be learnt. If the desired function is relatively simple it's a positive. However, if there are simpler features that are highly predictive; commonly named "shortcuts" or "spurious features", that are not present in the test environment, the SB can result in poor generalisation performance. Most existing works on mitigating the SB make various assumptions, either about the features present in the train and test domains or by assuming access to information about the test domain at train time. In this paper we review recent work on the SB and take a critical look at these assumptions.

## 1 INTRODUCTION

There are many examples of neural networks (NNs) trained with standard training techniques[1] using "shortcuts" (Geirhos et al., 2020) or spurious features[2] to make predictions instead of some desired attributes or "labelling function" (Puli et al., 2023; Kirichenko et al., 2023). For image classification tasks background, texture, and secondary objects are often considered spurious features even if predictive of the label Ye et al. (2024). For example Li et al. (2023) observed that many NNs trained on the ImageNet data set (Deng et al., 2009) use a text watermark, as a shortcut to classify images of boxes, rather than the pixels representing the boxes themselves.

The "Simplicity Bias" (SB) is normally attributed as the underlying factor causing models to use spurious features (Shah et al., 2020). The SB is the observation that there seems to be an implicit ordering over the possible mappings learnt by NNs with standard training techniques. These techniques are more likely to recover parameter settings encoding "simple features", Moreover, it has been shown that parameterisation corresponding to "simple functions" occupy the majority of the volume of the solution space[3] (Scimeca et al., 2021). However, there are many differing definitions over what features are "simple". Due to this implied preference ordering, if a data set contains multiple predictive features it is likely that the encoded function at test time will make most use of the "simplest" feature. Thus, any features of interest with high complexity may not be learnt during training, as simpler features are sufficient to minimise the loss. In this way simple features can be seen as distractors, that prevent the learning of more complex features. While, the collective understanding of this phenomena has steadily been improving, in this manuscript we take a critical look at existing works investigating the SB, "shortcuts" or "spurious correlation" and highlight some of the common assumptions often made by these works.

1. *Known Test Domain:* This is the assumption that one has fairly detailed knowledge of the test domain at train time, typically captured in the form of a "clean" data set.

2. *All Simple Features are Shortcuts* This is the assumption that all simple features are short cuts.

3. *Two Features Assumption:* This common simplifying assumption is only to consider data sets with two "features"; a desired feature to be learnt which is assumed to be complex and

---

[1] Such as Empirical risk minimisation (ERM) in combination with first order optimises such as SGD.

[2] Here we use "feature" to mean some characteristic of the input.

[3] The subset of parameter space which achieves low training loss

a short cut. The commonly used Water Birds data set has this structure. The bird is the desired feature to be learnt and the background as the short cut.

In this paper we discuss these assumptions and the implications of making them. Specifically we make the following contributions:

1. We highlight that the notion of "spurious features" relies on knowledge of the test domain, and give a characterisation of the different settings where spurious features can be problematic. We identify an under explored setting where the simplicity bias can cause poor generalisation performance. Specifically, learning in the presence of unknown domain shifts.

2. We show that due to typical data distributions living in an large ambient space the input sensitives of a data set does not necessarily correlate with conversional notation of simplicity or its difficulty in being learnt.

3. We investigate how SB affects learning on data sets where there are a hierarchy of features with different predictivities and availabilities. We show in this setting removing a specific unwanted feature, rarely results in an increase in performance on a different feature.

## 2 PRELIMINARIES

### 2.1 NOTATION

We use $\mathbb{D}$ to define a data set of $N$ training tuples $\mathbb{D} \triangleq \{(\boldsymbol{x}_n, \boldsymbol{y}_n)\}_{n=1}^N$, where $\boldsymbol{x} \in \mathbb{X} \in \mathbb{R}^d$, $\boldsymbol{y} \in \mathbb{Y} \in \mathbb{R}^q$. $\mathbb{X}$ and $\mathbb{Y}$ are the input and label manifolds and is $\mathbb{R}^d$ and $\mathbb{R}^q$ the ambient spaces of the input data and labels. $L(\boldsymbol{w}, \mathbb{D})$ is a loss function that takes as argument a vector of model parameters $\boldsymbol{w}$ evaluated over a data set $\mathbb{D}$. Finally, we denote $\mathbb{W}_{all}^*$ and $\boldsymbol{w}_{all}^*$ as the solution set and an element there of, of the following optimisation procedure.

$$\boldsymbol{w}_{all}^* \in \mathbb{W}_{all}^* \triangleq \underset{\boldsymbol{w}}{\arg\min} \ L(\boldsymbol{w}, \mathbb{D}_{all})$$

### 2.2 WHAT ARE FEATURES?

The description of the simplicity bias normally makes reference to "features", for clarity in this paper we use "features" to mean properties of a data set. To formalise this notion of features we find the definition presented in Ilyas et al. (2019) particularly compelling due to its flexibility. According to their definition a feature is some function of the input space to a single scalar that represents the presence or lack of the given attribute or concept. Formally:

$$\mathcal{F}_{all} = \{f : \mathbb{X} \to \mathbb{R}\}, \tag{1}$$

where $\mathcal{F}_{all}$ is the space of all features. For ease of discussion we will consider the problem of binary classification, with the label $y \in \{-1, 1\}$, except if stated otherwise. Thus following Hermann et al. (2024) we say a feature $f_j(\boldsymbol{x})$ has a predictivity of $\epsilon_j$ over a data set $\mathbb{D}$ if $\epsilon_j = P_{\{\boldsymbol{x},y\}\sim\mathbb{D}}(y = \text{sign}(f_j(\boldsymbol{x})))$. In words the predictivity of feature $f_j$ over a data set $\mathbb{D}$ is a measure the correlated with the label $y \in \{-1, 1\}$ and $\text{sign}(f_j(\boldsymbol{x}))$.

### 2.2.1 SIMPLE FEATURES

The simplicity bias is the observation that NNs are bias toward learning simple functions or features. This observation can be split into to parts, first that there seems to be a preference ordering over functions that is mostly transferable between models, and second this ordering seems to be correlated with the "simplicity" of the function. However, there is no consensus on the definition of "simple". Shah et al. (2020) define the simplicity of a function by the minimum number of linear pieces in the decision boundary that achieves optimal classification accuracy using that function. Unfortunately, this definition is not easy to quantify outside of toy data sets. Wang et al. (2023) suggest low frequency features are simple and hence learnt first. Vasudeva et al. (2024b) define simple as those that can be learn by models with fewer parameters. Conversely, Hermann & Lampinen (2020) look at what features convolutional neural networks (CNN's) learn first and suggest simple

features are those that can be most easily decoded by linear probing before training. Morwani et al. (2023) provide a precise definition for SB for 1-hidden layer neural networks, and suggest simple features can be characterised by a low dimensional projection of the inputs. More recently several works (Scimeca et al., 2021; Mingard et al., 2023) have suggested NN are biased toward learning Kolmogorov simple functions. All these metrics correspond the some measure of the information required in describing the function and thus In practice, outside of carefully constructed functions they are likely to be highly correlated.

## 2.3 IS INPUT SENSITIVITY A GOOD WAY TO MEASURE COMPLEXITY?

Recently Vasudeva et al. (2024a) investigated the simplicity bias of transformers. As transformers encode functions over sequences of discrete tokens the authors asked whether input sensitivity might offer a unified notion of simplicity? Their definition of input sensitivity measures how frequently the output of a function changes for a single token substitution of a given input, or set of inputs. Unlike the previously mentioned methods of measuring simplicity, this approach looks at a local property of the function around a given point or set of points, rather than global property. We note it is possible to design a data set which encodes a function with high input sensitivity but low complexity, or low input sensitivity but high complexity. In Section 2.3.1 we empirically back up this claim and show that input sensitivity does not always align with the preference order over functions. We suggest this quantity is more aligned with robustness than simplicity due to its obvious parallel to adversarial robustness defined over sequences of tokens. We note however, if a function has high sensitivity over the whole ambient space of which it is defined, it does follow that it must be highly complex. However, when sampling text, images or audio the data manifold only occupy a tiny subset of the ambient space, so for these data types input sensitivity does not necessary correlate with other defintion of "simplicity".

### 2.3.1 TOY COUNTER EXAMPLE

In this section we construct two families of binary classification data sets. For simplicity the input data in these data sets is sequences of two tokens "0" and "1". One data set has low complexity but high input sensitivity as defined in Vasudeva et al. (2024a) and one high complexity and low input sensitivity. The first is a linearly separable data set with each point adjacent to the decision boundary. This data set is very "simple", but has relatively high sensitivity. In contrast consider the repeated parity function, with each token repeated $2k + 1$ times. This data set has zero sensitivity for all perturbations of $k$ input tokens. Repeating each token gives a very simple way of reducing the sensitivity to token substitutions, however there exist far more sophisticated ways of encoding binary data that reduce its sensitivity to perturbations, as this problem has long been studied in the fields of error detecting and error correcting codes. Consider the data generating processes (DGP):

$$\boldsymbol{y}_n \sim \{0,1\}, \tag{2}$$

$$\boldsymbol{x}_n^{simple} \sim \begin{cases} \text{if } \boldsymbol{y}_n = 0, \{\boldsymbol{s} \in \{0,1\}^l : \boldsymbol{s} \in \text{perm}([1,...,1,0,0,\ldots,0,])\} \\ \text{if } \boldsymbol{y}_n = 1, \{\boldsymbol{s} \in \{0,1\}^l : \boldsymbol{s} \in \text{perm}([1,...,1,1,0,\ldots,0,])\} \end{cases}, \tag{3}$$

$$\mathbb{D}_{simple} \triangleq \{(\boldsymbol{x}_n^{simple}, \boldsymbol{y}_n)\}_{n=1}^N. \tag{4}$$

Where perm($\boldsymbol{b}$) is the set of all permutations of a sequence $\boldsymbol{b} = [b_1, b_2, \ldots, b_l]$. Note how this data set has two hyper-parameters $l$ (the length of $\boldsymbol{x}$) and $k_{simple}$ that controls the number of 1's in each example. Specifically, $\boldsymbol{x}_i$'s with $y_i = 0$ contain $k_{simple}$ "1" tokens and $\boldsymbol{x}_j$'s with $y_j = 0$ contain $(k_{simple} + 1)$ "1" tokens. Due to the difference in number of "1" tokens when considering this data as a point on the $l$ dimensional unit cube it is linearly separable.

$$\boldsymbol{y}_n \sim \{0,1\}, \tag{5}$$

$$\boldsymbol{a}_n \sim \begin{cases} \text{if } \boldsymbol{y}_n = 0, \{\boldsymbol{s} \in \{0,1\}^d : \boldsymbol{s}^\top \mathbf{1} \bmod 2 = 0\} \\ \text{if } \boldsymbol{y}_n = 1, \{\boldsymbol{s} \in \{0,1\}^d : \boldsymbol{s}^\top \mathbf{1} \bmod 2 = 1\} \end{cases}, \tag{6}$$

$$\boldsymbol{x}_n^{complex} = [\boldsymbol{a}_n, \ldots, \boldsymbol{a}_n], \tag{7}$$

$$\mathbb{D}_{complex} \triangleq \{(\boldsymbol{x}_n^{complex}, \boldsymbol{y}_n)\}_{n=1}^N \tag{8}$$

Again this data set has two hyper-parameters $d$ the length of $\boldsymbol{a}$, and $k_{complex}$ which controls the number of repeats of $\boldsymbol{a}_n$ in each vector $\boldsymbol{x}$, (Equation 7). The length of each $\boldsymbol{x}_n^{complex}$ $l = k_{complex}\dot{d}$,

but we see the lenght to match the "simple" data set $\mathbb{D}_{simple}$. Note how for $k \geq 1$ a $k$ bit substitution will never swap the class and thus this data set has zero input sensitivity as defined in Vasudeva et al. (2024a).

Finally we introduce a third data set $\mathbb{D}_{both}$ where $\boldsymbol{x}_n^{both} = [\boldsymbol{x}_n^{complex}, \boldsymbol{x}_n^{simple}]$, or $\boldsymbol{x}_n^{both} = [\boldsymbol{x}_n^{simple}, \boldsymbol{x}_n^{complex}]$ in words the sequences are constructed by concatenating one sequence from each data set with the corresponding output token. Hence, we have a data set with two predictive features, that could be used to predict the label. For all data sets we split the examples into test and train subsets and Train a mini-GPT model for 10,000 iterations for various setting of $k$ and $l$.

**Results**  The simple data set with $l < 1000$ and any value of $k_{simple}$ is easy to learn and the model achieves test error rates 1%. Conversely, the complex data set is already too difficult for the model to learn with $l = 48, k_{complex} = 3$, with test error rates 50%. Finally, we assess the performance of models trained on $\mathbb{D}_{both}$ on data sets where one of the features has been swapped to be predictive of the other class, this lets us see which feature is used buy the model for prediction. Here, for most values of $l, k_{complex}, k_{simple}$ the model learns to predict purely based on the $\boldsymbol{x}_n^{simple}$. In this controlled experiment we have shown that classic metrics of feature simplicity are more predictive of ease in learning features when compared to input sensitivity.

## 3   WHAT ARE SPURIOUS FEATURES?

The consensus is spurious features are those that are predictive of the label in the train set but fail to be predictive at test time. We claim the concept of a feature being spurious inherently relies on additional side information of the test domain, whether this be agreement with a human annotator or access to additional data deemed appropriate by a practitioner. For example, (Geirhos et al., 2020) suggest a shortcut performs well on standard benchmarks but fails to transfer to more challenging testing conditions, such as real-world scenarios. This definition relies on having knowledge of the more challenging testing condition, which is typically captured in the form of a "clean" data set or as human knowledge about likely domain shifts. Without extra information at training time it is impossible to know which features are spurious and which are not. One only has a data set containing a continuum of different features with different predictive powers.

### 3.1   DO BLAME DISTRIBUTION SHIFT!

(Puli et al., 2023) suggested distribution shift should not be blamed for the problems cause by short-cut learning. To back up this claim they provide theoretical results showing short cut learning for a linear binary classification problem even when the same data generating process (DGP) is used to generate both train and test domains. However, their theoretical results rely on the sub-sampling of the DGP during training, and only hold for bounded data set sizes. Thus in their theoretical results they actually have test and train data sets with slightly different distributions. Additionally Statistical Learning Theory (Vapnik, 1999), suggests for convex problems like the one considered Empirical risk minimisation should return a model that minimises the risk over the training distribution. If one knows *a priori* there will be no distribution shift at test time, and if the train set is large enough to distinguish noise from signal, one can learn any feature as they will still be predictive. Thus, spurious features are only problematic in combination with domain shift. However, for real world training sets there is likely to always be a shift in distribution between test and train domains, either due to sub sampling or differences in how the data were gathered (Ye et al., 2024).

### 3.2   WHERE ARE SPURIOUS FEATURES ACTUALLY A PROBLEM?

We now discuss a number of settings where spurious features can lead to poor generalisation. We divide up these settings based on the information present at train time about the test distribution.

**A Data Problem**  This setting is characterised by having limited data of the desired test environment, commonly caused by a high cost in gathering or approximating representative test data. If large quantities of appropriate test data were available at train time, one would simply train on this data instead. Thus, at train time one knows exactly what domain shift is present between the test and train environments. This setting is also obtained after a model has been deployed, and been found to be failing on the test data. This problem can be viewed as a data problem and if possible boot

strapping more appropriate test data can offer a solution. Outside of gathering more data a number of good general purpose approaches have been proposed. For example last layer training methods as discussed in Section 5 typically make the assumption of asseess to a small amount of test data.

**An Alignment Problem**    This setting is characterised by one having side information in the form of a prior belief about the test distribution, or a given feature of interest. One may or may not have a mathematical understanding on the form of function that is desired to be learnt, such as a low sensitivity to certain transformations of the data. None the less one is trying to train an NN to predict in a manner deemed appropriate by one or more humans supervisors (Xiao et al., 2021). This setting requires a human in the loop determining, and adjusting, the training procedure to try and promote invariance to certain features if they are being used for prediction. This process is hard to automate and expensive. In the worst case a spurious feature maybe identified, removed from the training data, and the model retrained, only for the model to learn to use another undesirable feature for prediction. This can lead to a metaphorical game of "Whack-A-Mole" as described in Li et al. (2023).

**Limited Knowledge of the Test Domain**    While the above two settings are well studied they both make a fairly big assumption. That is, they assume detailed knowledge about the test domain at train time. While this assumption is likely true in the majority of use cases we claim there are setting where it does not. For concrete example a practitioner would likely not know what types of shifts to expect in bird or whale song when moving between region or species. One likely has a poor understanding of which features of these data set are likely to change at test time and which are not. Other examples indicative of this setting include privatised tabular data where column headings have been removed.

This setting where one has very limited knowledge of the deployment domain remains under explored. In order to make learning feasible in this setting one would need to assume some commonality between, test and train domains. One way to ensure learning on the train domain translates to the test domain would be to assume and unknown subset of the features in the training data set are present at test time. In this setting the best course of action is to learn a model sensitive to a large number of diverse features (Teney et al., 2022a;b). However, once we have knowledge of a test domain it is possible to conclude what features were good and thus it is difficult to effectively simulate this lack of knowledge. To address this difficulty one option would be to evaluate the performance after a variety of domain shifts, and report the average. While the works of Teney et al. (2022a;b) considered training an ensemble of models that are predictive to multiple features for best result they also assume access to test data for model selection. In the next subsetion we formalise this under-explored learning problem.

**Learning Problem**    In order to formalise the problem of learning where one has very limited knowledge of the deployment domain. We consider settings when the sets of features present in the train and test distributions obey the following hierarchy:

$$\mathcal{F}_{test} \subseteq \mathcal{F}_{train} \subseteq \mathcal{F}_{all}. \tag{9}$$

In other words we assume that there is a subset of predictive features at train time $\mathcal{F}_{train} = \{f_1, \ldots, f_J\}$ and that any distribution shifts between train and test simply remove some of the features present at train time, however we do not know which until test time. Thus, we want to ensure the model has high generalisation performance on *any* subset of the features. While only accessing $\mathbb{D}_{all}$ at train time the learning problem is to find a model that minimises:

$$\boldsymbol{w}^{**} = \text{argmin}_{\boldsymbol{w}} \sum_{j \in \mathcal{F}_{train}} \mathcal{L}(\boldsymbol{w}, \mathbb{D}_j), \tag{10}$$

where $\mathbb{D}_j$ is a data set that only contains a subset of the features present in the $\mathcal{F}_{train}$, and all other features are not present.[4]

---

[4]The loss described in (10) is a multi-objective optimisation and while we have shown the reduction of the losses using a sum, a different function to control the trade off between the different objectives could be more suitable in some scenarios.

| Dataset | Task-relevantinvariant feature | Surrogate/spurious feature |
|---------|-------------------------------|----------------------------|
| Waterbirds | Bird type (waterbird/landbird) | Background (water/land) |
| CelebA | Hair colour (blonde/other) | Gender (female/male) |
| MultiNLI | Reasoning | Negation words |
| CivilComments-WILDS | Sentiment (toxic/non-toxic) | Race, Gender, Religion) |
| Colored-MNIST | Digit ($< 5$ or $\geq 5$) | Color (red/green) |
| $Camelyon_{17} - WILDS$ | Diagnoses (tumor/no tumor) | Hospital |
| Adult-Confounded | Income ($< \$50k$ or $\geq \$50k$) | Race, Gender |

Table 1: Summary of the datasets we consider. Spurious features seem simpler than invariant features Vasudeva et al. (2024b)

### 3.2.1 TWO FEATURES ASSUMPTION

The next prevalent assumption we want to highlight is what we call the "two feature assumption". Specifically, that is assuming data sets only contain two features a complex desired feature and a simple spurious feature. Table 1 is adapted from Vasudeva et al. (2024b) and highlights how frequent this assumption is in common simplest bias data sets.

We conjecture this assumption is made typically for the following few reasons. As the number of features increases the number of unique feature combination grows exponentially and hence, to maintain the same number of training points for each feature combination one requires an exponential amount of data. Additionally, labelling real world data sets with multiple features is expensive. Thus, the few data image sets that do come with multiple feature annotations typically make use of synthetic or composite images (Lynch et al., 2023; Li et al., 2023). To the best our knowledge all spurious correlation NLP data sets still make the two feature assumption.

The main issue with the two feature assumption is for data sets which have it by design, promoting invariance to the simplest feature typically results in learning the other complex feature. However, for data sets with any more than two features this is no longer true, and not learning the simplest feature does not ensure the most complex is learnt. This focus on two feature data sets has lead to methods like those discussed in Section 5 which focus more on not learning simple features, rather than either learning i) multiple features or ii) the features with highest predictivity.

In the next section we construct data sets with multiple features and show removing or not learning a feature very rarely results in a specific more predictive feature being learnt.

## 4    BINARY DATA SET EXPERIMENTS

In this section we introduce a family of toy data sets for exploring the simplicity bias of NNs. These data sets are chosen so that we have full control over the features present. The data sets are constructed by concatenating together several binary features with different availabilities and predictivities (Hermann et al., 2024). We construct a data set that has a hierarchy of features where each feature has a greater predictivity but lower availability than the previous.

**Binary Features.**  If we denote $\mathbb{B}_a \triangleq \{-1, 1\}^a$ then we define a binary feature $f_j$ as a mapping $f_j : \mathbb{B}_a \to \{0, 1\}$. We design these feature or function so the label is the parity bit of the input. Specifically, when a feature $\boldsymbol{z} \in \mathbb{B}_a$ is predictive of label[5] $y \in \{0, 1\}$, the following relation will hold:

$$\sum_{i=1}^{a} \mathbf{1}_{[z_i=1]} \bmod 2 = y$$

However, by design only for $\epsilon$ of the training examples is the feature predictive of the label $P(\sum_i \mathbf{1}_{[x_i=1]} \bmod 2 = y) = \epsilon$. This is done by introducing feature noise, swapping the feature to be predictive of the other class. Binary features of this form are non-linear for all features with more than two bits $a > 1$. A feature of this form with $a = 2$ encodes a XNOR relation. For larger $a$ these features match the parity learning set up of $k$ bits as described in (Raz, 2018; Daniely

---

[5]By predictive of the label we mean not subject to feature noise.

| Feature name | Dimensional (Bits) | Availability | Predictivity $\epsilon$ |
|:---:|:---:|:---:|:---:|
| $f_1$ | 1 | very high | 60% |
| $f_2$ | 2 | high | 70% |
| $f_3$ | 4 | medium | 80% |
| $f_4$ | 8 | low | 90% |
| $f_5$ | 16 | very low | 100% |
| $n_1$ | 48 | N/A | 100% |

Table 2: The set of features present in Data Set $\mathbb{D}_{all}$. A feature $f_i$ is only predictive if all features $f_j$ where $j > i$ are predictive also. Hence for 10% of the data set only $f_5$ is predictive. The noise feature while randomly sampled can be fully predictive by noise pattern forming a unique identifier for each sample that the network memorises.

& Malach, 2020). As the demensionality of the feature increases it becomes less available and thus the difficultly in learning increases.

**Data Generating Process**  We construct a data set $\mathbb{D}_{all}$ by first deciding on a set of features $\mathbb{J} \triangleq \{f_1, f_2, ..., f_J\}$ and each features corresponding availability and predictive powers $\epsilon$. For example Table 2 shows the version of this data set we focus on in this paper containing five features with increasing complexities and predictive power, its also contains a noise feature $\{n_1\}$ which is randomly sampled from $\mathbb{B}_a$. We pick a data set size, $n = 2^{16}$ and create an equal number of samples for each class $y_n \in \{0, 1\}$. Each $\boldsymbol{x}_n$ is constructed by first sampling vectors $\boldsymbol{z}_{n,j} \in \mathbb{B}_{a_j}$. The features are sampled to ensure that a feature $f_j(\boldsymbol{z}_{n,j}) \neq y_n$ if any $f_k(\boldsymbol{z}_{n,j}) \neq y_n$ where $f_k$ has higher predictive power that $f_j$ or $\epsilon_k > \epsilon_j$. In words, we ensure 10% of the data can only be correctly classified using $f_5$, there is another non-overlapping 10% that can only be correctly predicted using $f_4$ or $f_5$ and so on. Thus the features form a hierarchy, and this allows us to have a data set with many features, but ensure the only way to achieve high performance on all samples is learn the feature with the highest predictive power, or conversely memorise this 10% of examples. Once the vectors $\boldsymbol{z}_{n,j}$ have been sampled so the desired properties of the features in $\mathbb{J}$ hold, we then lift each $\boldsymbol{z}_{n,j}$ feature into an ambient space of $\boldsymbol{z}'_{n,j} \in \{-1, 0, 1\}^{16}$ by padding zeros and then concatenating together $\boldsymbol{x}_i = \{[\boldsymbol{z}'_{i,1}, \boldsymbol{z}'_{i,2}, \ldots, \boldsymbol{z}'_{i,|\mathbb{J}|}]\} \in \{-1, 0, 1\}^{128}$, for each $y_i \in \{0, 1\}$. Lastly, we define a masking function $m_{\mathbb{M}}(\boldsymbol{x}), \forall \mathbb{M} \subseteq \mathbb{J}$ that zeros out all features not in $\mathbb{M}$. This data set then has the property that $P_{n \sim N}(f_j(m_j(\boldsymbol{z}_{n,j})) = y_n) = \epsilon_j$. While this data set is small and synthetic it allows understanding to be gained about learning in the presence of multiple different features, where we have price control.

**Training Procedure**  We train an MLP with 165k parameters and 4 layers and a constant width of 256 on $\mathbb{D}_{all}$. This experiment is repeated with only a certain subset of the features present throughout training. Where features are not present they are masked out by a zeros. We additionally include 3 stochastic masking schemes throughout training "rand","rand all" and 'subset'. "rand" randomly samples one features to use for each iteration. "rand all" also includes all features (or no mask) in the possible features to sample. "subset" samples are random subset of the features at each iteration. We run 5 repeats and train with Adam for 200 epochs.

**Results**  The mean performance across runs is detailed in Table 3. From the settings where only one feature was present we can observe that the model is more than capable of learning any of the solitary features. Note how the noise feature is the only one which fails to generalise to the test domain. From the runs where a single feature and the noise features were present we can observe that even the present of noise can act as a short cut. From the middle block of results in 1 we can see the model continues to learn the harder but more predictive features as they are added to the training set up to a point, but it struggles to learn the least available in the presence of the more available less predictive features. The penultimate set of runs, shows the performance when the simplest feature is repeatedly removed. We can observe that only when all four simpler features have been removed from the input does the model learn the most predictive feature $f_5$. This suggest techniques that aim to remove the more simplistic features will likely struggle to learn more complex features. Finally for the stochastic masking schemes, particular "rand all", we can see that the model is actually capable of learning all the features, individually but also achieve good test when all features are

| Train Features | train | $f_1$ | $f_2$ | $f_3$ | $f_4$ | $f_5$ | $n_1$ | all | Avg |
|---|---|---|---|---|---|---|---|---|---|
| $\{f_1\}$ | 60.0 | 59.7 | 56.0 | 49.7 | 49.3 | 50.7 | 50.0 | 56.0 | 53.0 |
| $\{f_2\}$ | 70.0 | 50.0 | 69.7 | 53.7 | 48.7 | 49.7 | 49.3 | 52.3 | 53.3 |
| $\{f_3\}$ | 79.7 | 49.3 | 49.3 | 80.0 | 50.0 | 49.7 | 50.0 | 55.3 | 54.8 |
| $\{f_4\}$ | 90.0 | 50.0 | 43.0 | 51.0 | 90.0 | 50.3 | 50.0 | 75.7 | 58.6 |
| $\{f_5\}$ | 100.0 | 50.0 | 50.0 | 51.7 | 49.3 | 100.0 | 50.0 | 89.0 | 62.9 |
| $\{n_1\}$ | 99.7 | 49.3 | 49.3 | 49.3 | 49.3 | 49.7 | 50.7 | 50.7 | 49.8 |
| $\{f_1, n_1\}$ | 99.7 | 50.0 | 50.0 | 50.0 | 50.0 | 50.0 | 50.3 | 53.0 | 50.5 |
| $\{f_2, n_1\}$ | 99.3 | 50.0 | 63.7 | 50.0 | 50.0 | 50.0 | 49.3 | 59.0 | 53.1 |
| $\{f_3, n_1\}$ | 99.7 | 50.0 | 50.0 | 51.3 | 50.0 | 50.0 | 50.0 | 70.0 | 53.0 |
| $\{f_4, n_1\}$ | 99.3 | 50.0 | 50.0 | 50.0 | 51.7 | 50.0 | 50.7 | 50.0 | 50.3 |
| $\{f_5, n_1\}$ | 100.0 | 50.0 | 50.0 | 50.0 | 50.0 | 50.0 | 50.3 | 50.0 | 50.0 |
| $\{f_1\}$ | 60.0 | 59.7 | 56.0 | 49.7 | 49.3 | 50.7 | 50.0 | 56.0 | 53.0 |
| $\{f_1, f_2\}$ | 70.0 | 59.7 | 69.7 | 50.3 | 50.0 | 50.0 | 49.7 | 48.0 | 53.9 |
| $\{f_1, f_2, f_3\}$ | 79.7 | 59.7 | 47.3 | 79.0 | 50.3 | 50.0 | 49.7 | 54.3 | 55.8 |
| $\{f_1, f_2, f_3, f_4\}$ | 90.0 | 43.0 | 56.7 | 53.7 | 56.0 | 49.7 | 49.0 | 87.3 | 56.5 |
| $\{f_1, f_2, f_3, f_4, f_5\}$ | 99.7 | 50.0 | 63.0 | 66.0 | 48.3 | 50.0 | 50.0 | 70.7 | 56.9 |
| $\{f_5\}$ | 100.0 | 50.0 | 50.0 | 51.7 | 49.3 | 100.0 | 50.0 | 89.0 | 62.9 |
| $\{f_5, f_4\}$ | 99.7 | 50.0 | 50.0 | 50.0 | 47.7 | 45.3 | 49.7 | 84.0 | 53.8 |
| $\{f_5, f_4, f_3\}$ | 99.0 | 50.0 | 50.0 | 56.0 | 49.7 | 49.3 | 49.7 | 68.0 | 53.2 |
| $\{f_5, f_4, f_3, f_2\}$ | 99.3 | 49.3 | 53.3 | 63.0 | 49.0 | 50.3 | 49.7 | 69.0 | 54.8 |
| $\{f_5, f_4, f_3, f_2, f_1\}$ | 99.0 | 46.7 | 63.3 | 53.0 | 49.0 | 50.3 | 50.0 | 70.7 | 54.7 |
| $\{\text{rand}\}$ | 81.3 | 59.7 | 69.7 | 80.0 | 90.0 | 100.0 | 50.0 | 49.7 | 71.3 |
| $\{\text{rand all}\}$ | 83.3 | 59.7 | 69.7 | 80.0 | 90.0 | 100.0 | 49.7 | 84.0 | 76.1 |
| $\{\text{subset}\}$ | 87.7 | 59.7 | 69.7 | 80.0 | 90.0 | 49.7 | 50.3 | 76.7 | 68.0 |

Table 3: Results for a MLP model trained on different subset of features present in $\mathbb{D}_{all}$. The other feature are excluded via masking. We provide the accuracy of the model evaluated on all features. Note that "rand", and "rand all" and "subset" correspond to different stochastic masking schemes of the features applied during training.

present at once. While this toy setting is contrived and hardly realistic of real world data. It does highlight the issues with trying to learn a certain feature such as $f_5$, by trying to not learn other features. Note that $f_5$ is the only feature that outside of the noise feature that leads to 100% training accuracy.

## 5 RELATED WORK

**Simplicity Bias** It's hard to pin point the exact origin of the concept of the SB, however in their work Learning Qualitatively Diverse and Interpretable Rules for Classification, Ross et al. (2018) introduced a way to identify a maximal set of distinct but accurate models for a dataset. The authors demonstrated empirically that, in situations where the data supports multiple accurate classifiers, SGD tends to recover simpler, more interpretable classifiers rather than more complex ones. Valle-Perez et al. (2019) suggest the good generalisation performance of neural networks was because the parameter-function map is biased towards simple functions. Nakkiran et al. (2019) suggested when using SGD to train a NN, the NN learns functions of increasing complexity. Starting by first learning a linear model which is retained even to convergence. Scimeca et al. (2021) show that solutions corresponding to Kolmogorov-simple cues are abundant in the parameter space and are thus more likely found by DNNs. More recently, Bell & Sagun (2022) showed that SB can lead to larger performance disparities between different groups of training data. Finally, Mingard et al. (2023) claim NN have an inbuilt Occam's razor. Their analysis also suggests a strong bias to functions with low Kolmogorov complexity. Shah et al. (2020) was the first work to focus on the negative phenomena that can result form the SB. Specifically, they showed that i) SB can be extreme, causing networks to rely solely on the simplest feature, (ii) extreme SB exacerbates performance degradation from distribution shifts and adversarial attacks, (iii) SB can hinder generalisation even when simpler features are less predictive, and (iv) common strategies like ensembles and adversarial training struggle to mitigate SB's drawbacks. While the observation that neural networks were

biased to learning simpler functions was known before this, most works considered it to be a positive property. A number of papers have studied the SB from a more theoretical perspective. Soudry et al. (2018); Morwani et al. (2023) have shown that minimising ERM finds a maximum margin model, if one exists. While this implicit bias may be desirable in some cases it (Puli et al., 2023) showed that it prevents linear models learning to use the label when it is part of the input and suggest a number of losses to mitigate this phenomena.

**Mitigating the Simplicity Bias and Spurious Correlations**   A number of works have presented algorithms to help overcome the problem of learning, in the presence of spurious correlations. Learning de-biased representations with biased representations (Bahng et al., 2020), suggest it is much easier to define a set of biased representations than to define and quantify bias. Thus they find a de-biased representation by encouraging it to be different from a set of representations that are biased by design. Nam et al. (2020) present a similar method where they also purposely train a network designed to amplify biases. This network is then used to debias the training of a second network by focusing on samples that go against the biased network. Tiwari & Shenoy (2023) suggest a related but different approach that they call feature sieve. They show that simple features can often be found in the earlier layers of a neural network. They use an auxiliary network to alternately identify predictive features and erase them only at the lower network layers of an unbiased network, thereby allowing the higher network levels to extract and utilise richer, more meaningful representations. While these works all report compelling results on data sets that are biased by design, these data sets are chosen to include simple distractors and make the assumption the features of interest are not "simple". Finally, these approaches often have a large number of hyperparameters that are required to tune for each setting, without access to a unbiased data set to select these on its not clear how well these methods work in practice.

**Last Layer Retraining**   A number of works Rosenfeld et al. (2022); Kirichenko et al. (2023); Izmailov et al. (2022) suggest for many large scale real world data sets SGD & ERM already finds models whose last layer activations encode a broad range of features. They show these features are sufficient for out-of-distribution (OOD) generalisation by achieving excellent performance when only retraining the last layer of a model on the new domain. Their results suggest that doing well on problem (10) corresponds to doing well at the problem of robust classification on top of a sufficiently powerful feature extraction back bone. Izmailov et al. (2022) assess the relative contributions of i) the representation produced by the feature extractor and ii) the final classifier. They show that many techniques such as early stopping and strong weight decay can improve the worst group accuracy by learning a better classifier, but do not lead to a consistent improvement in terms of the quality of the learned feature representations. Conversely, the quality of the feature representations depends heavily on the base model architecture and pre-training strategy. However, these works all assume access to a "clean" data set drawn from the test domain which make them best suited to the setting where gathering data from this domain is possible but expensive while gathering data from a related simpler domain is cheap. This is not the setting we consider, as we assume no access to the test domain at train time. Addepalli et al. (2023) suggest that ensuring the last layer activation (or learnt features) can be reconstructed from the logits is a good approach for learning a robust classifier that mitigates the simplicity bias. While this approach encourages the logits, and hence the prediction to encode more of the information from the learnt features, this method seems to unfortunately require a large number of classes to achieved good results for classification tasks, limiting its applicability.

# 6   CONCLUSIONS

In this work we have reviewed many of the recent papers discussing the simplicity bias and short cut learning. We have highlighted a number of common assumption that likely need to be relaxed in future work in order to ensure our understanding of this phenomena continues to improve. We have also addressed a few claims about the Simplicity bias made by recent work that we believe are not accurate. Finally, we have introduced a toy example for studying the behaviour of NN on data sets that contain a large number of features.

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
