# OpenReview forum: "Questioning Simplicity Bias Assumptions"
_ICLR.cc/2025/Conference — ICLR 2025 Conference Withdrawn Submission_

### Official Review · Reviewer_YdxK · 2024-10-23

**Soundness:** 2
**Presentation:** 2
**Contribution:** 1
**Rating:** 1
**Confidence:** 4

**Summary:**

The paper reviews different works about simplicity bias, and discuss several assumptions often made when studying simplicity bias, and how relevant they are in practice. The paper introduces a toy-problem showing how the presence of common features which are less predictive can mask the learning of less common features which are more predictive.

**Strengths:**

The paper discusses simplicity bias in a clear way, and it is easy to read. When discussing the different assumptions made by other papers, the paper is very extensive, and shows large various of examples from the literature for each point.

**Weaknesses:**

The paper in its current form leans heavily towards being a review or survey, offering limited new experimental or theoretical contributions. Many of the claims and assertions are inadequately supported, lacking both empirical evidence and theoretical depth.

For instance, when discussing assumptions made by works on simplicity bias, the paper critiques these assumptions but provides little to no empirical or theoretical evidence to support these critiques. A stronger approach would include relevant models and datasets that are commonly used in the study of deep learning to substantiate the claims.

Take, for example, the discussion of the "Two Features Assumption." Rather than simply listing papers and datasets that operate under this assumption, I would expect the paper to include experiments with deep networks on real datasets or a theoretical framework demonstrating that avoiding the learning of one "bad" feature does not necessarily facilitate the learning of the "good" feature. This is particularly important in real-world datasets, where what appears to be a binary feature distinction ("birds" vs. "background") may actually encompass a complex set of features -- as both birds and backgrounds are complex objects which contain many features themselves.

Another example is the distinction made between "spurious features" and domain shift. The paper references several works that discuss this point but does not add any novel insights. It is self-evident that discrepancies between training and test set performance must involve some degree of distribution shift -- otherwise the performance would be identical. The paper needs to clarify what original contributions are being made beyond this well-established fact.

Overall, the paper reads more like a survey, and while survey papers can have value, they are typically accepted at top-tier conferences like ICLR only if they offer original insights grounded in strong evidence. This paper, however, does not meet that criterion.

**Questions:**

not relevant

---

### Official Review · Reviewer_bzLM · 2024-11-03

**Soundness:** 1
**Presentation:** 2
**Contribution:** 2
**Rating:** 3
**Confidence:** 3

**Summary:**

The paper surveys theoretical literatures around: simplicity bias ~= spurious correlation ~= shortcut learning

They propose 3 contributions: (i) conceptual notion that spurious features depend on the test domain; (ii) questions around the definition of simplicity; (iii) analysis in a hierarchical setting;


Sec2 run synthetic simulation on a binary example.
Sec3 discussions papers around distribution shift + spurious correlation.
Sec4 performs the hierarchical experiments, on MLP

**Strengths:**

* Visits an important problem.
* Have two sets of experiments.

**Weaknesses:**

* "the notion of “spurious features” relies on knowledge of the test domain” is not a new message to the community.
* Lacks experiments with real datasets and practical neural network architecture.
* Also lacks theoretical framework to support the authors claim.

**Questions:**

* L66  "conversational" should be "conventional"?

---

### Official Review · Reviewer_CAJH · 2024-11-03

**Soundness:** 2
**Presentation:** 3
**Contribution:** 2
**Rating:** 3
**Confidence:** 5

**Summary:**

The paper surveys works addressing the phenomenon of simplicity bias in deep neural networks. The paper critically examines the assumptions of past work in defining features as spurious, and surveys the various definitions of “simple features” provided in previous work. The paper then emphasizes that previous works conflate spurious features and distribution shifts, arguing that the former is a problem only under the latter settings. More fundamentally, determining that feature to be spurious needs knowledge of or assumptions on the test domain. An approach of previous works with minimal assumptions is to learn as many features of the data from the train domain as possible, which is formalized by the authors here. However, the paper presents a counter example showing that learning a more complex feature is difficult unless all simpler features are discarded. Finally, the work surveys existing methods to combat simplicity bias and discuss how their assumptions match up with the rest of the paper.

**Strengths:**

1. The paper does a good job of surveying the field of simplicity bias, with the literature review being fairly comprehensive
2. The paper presents various definitions of simplicity, and tests them for their usefulness well .
3. The experiments and hypotheses arguing against the two features hypothesis are nice, showing that SB mitigation methods might not scale to real world data.

**Weaknesses:**

**Contributions -**
1. “We identify that simplicity bias in the presence of unknown distribution shifts can cause poor generalization performance”. This paper is not the first to propose this. E.g. Adepalli et al 2023 also consider the task of learning in the presence of unknown distribution shifts (domain generalization), and show that reducing some notion of simplicity bias can lead to better generalization performance.
2. Two features assumption - This assumption is a limitation of the datasets. However, there are several works which do not make this assumption which are not mentioned by  the paper (Singla & Feizi 2021, Moayeri et al 2023), which aim to detect multiple spurious correlations in large scale datasets, going beyond the two features model.
3. The paper proposes a counter-example for input sensitivity as a definition of simplicity, however, it ignores other definitions?

**Empirical evidence -**
1. The counter-examples in the paper seem to be very contrived. While this is acknowledged in the paper, it is not clear if these counter examples have much resemblance with real world data. A short discussion of why such counter examples might be practically relevant would benefit the paper greatly.
2. For the second counter-example, I would have liked to see how some of the SB mitigation methods actually perform, i.e. while ERM cannot learn the complex feature unless all simple features have been eliminated from the data, can any of the current methods learn $f_5$? If that is the case, then the claims of this paper might weaken.
3. I would have liked to see some recommendations by the authors on future papers in this field. E.g., a better definition of simple features, a recommendation for a dataset away from the two feature assumption etc.

**Questions:**

1. Do the authors have any recommendations for SB papers wrt each assumption they have put forth?
2. Can the authors comment on how representative SB mitigation methods fare for their counter examples?

---

### Official Review · Reviewer_Eq5d · 2024-11-05

**Soundness:** 2
**Presentation:** 2
**Contribution:** 2
**Rating:** 3
**Confidence:** 4

**Summary:**

This paper examines key assumptions in recent work on simplicity bias and its connections to shortcut learning. The analysis focuses on three assumptions in the literature: (1) the test distribution is known during training, (2) simple features inherently act as shortcuts, and (3) datasets can be decomposed into binary "core" versus "spurious" features. The paper argues that while these assumptions have enabled theoretical analysis of simplicity bias in controlled settings, they may limit our understanding of how models learn features in real-world scenarios.

**Strengths:**

- The paper shows that input sensitivity alone is not always a good proxy for simplicity bias. One can construct examples where a model has (a) low complexity but (b) high input sensitivity (Section 2.3.1)
- Introduces a toy dataset comprising many features to analyze simplicity bias in a controlled setting

**Weaknesses:**

- The paper correctly identifies the lack of consensus on what constitutes a "simple" feature and critiques definitions from prior work as either difficult to quantify in non-toy settings or poorly motivated. While this is a valid concern that has impacted research on simplicity bias in real-world datasets, the same critique applies to their chosen definition of "features." The definition adopted from Ilyas et al. - any function that maps input to a scalar value - is so general that it provides limited practical insight into neural network feature learning. However, the lack of precise definitions for "simple features" or "features" need not prevent meaningful investigation of simplicity bias.
- Section 3.1 presents as novel the observation that with sufficient training data and no distribution shift at test time, models can learn any predictive feature. This is neither surprising nor a blind spot in existing literature. Prior work on simplicity bias and shortcut learning specifically constructs OOD test sets with and without spurious features to study how models rely on such features.
- Section 3.2's finding that spurious features can lead to suboptimal in-distribution performance has been previously demonstrated, for instance in Section 5 of https://arxiv.org/abs/2006.07710.
- The paper overstates the prevalence of the two-features assumption in prior work. Experiments in https://arxiv.org/abs/2006.12433 and https://arxiv.org/abs/2006.07710 demonstrate simplicity bias with features having varying levels of predictivity and simplicity. Even in seemingly binary settings like waterbirds, model biases manifest at fine-grained subpopulation levels (https://arxiv.org/abs/2211.12491).
- The statement "However, for data sets with any more than two features this is no longer true, and not learning the simplest feature does not ensure the most complex is learnt" oversimplifies the relationship between dataset size and feature learning. Whether a model learns complex features or overfits to training data depends critically on dataset size, given that it doesn't learn the "simplest" feature.

**Questions:**

In terms of scope and contribution, this work would be better positioned as a review paper or blog post examining assumptions in simplicity bias research. The paper lacks the focused problem statement and specific research questions expected in a research track submission. While it correctly observes that real-world data rarely presents clear distinctions between spurious and core features, this insight doesn't diminish the value of simplified assumptions for studying underlying mechanisms in controlled settings. I would like to know what the main technical contribution of this work a bit better.

---

### Public Comment · ~Damien_Teney1 · 2024-11-16
**Public comment; nice survey**

Question from an external reader here (not a reviewer):

In Section 2.3.1 ("Toy Counter Example"), two datasets are introduced with low complexity/high sensistivity, and vice versa. What is the defintion of "simplicity" used? Section 2.2.1 precisely discusses the fact that there are many different defintions, so I cannot figure which one the authors use.

FWIW I found this paper to be a nice survey of common limitations in the existing literature on debiasing/shortcut learning/etc. It's a paper I would like to see published at some point so that I could cite it when refering to these issues in my own writings. But like the reviewers pointed out, ICLR is probably not the right venue. The paper also has a ton of typos and would benefit from a lot of editing to make the reading easier. I hope this comment is helpful to the authors!

---

> ### Author Response · Authors · 2024-11-17
> **Simplicity Definition**
>
> Thank you for your kind words.
>
> In answer to your question in Section 2.2.1 the metric where this is most self evident is the one from Shan et al. 2020 where complexity is defined as the"minimum number of pieces required by a piecewise linear classifier to attain optimal accuracy".

---

### Author Response · Authors · 2024-11-23
**Thank you**

We thank all the reviewers for their thoughtful and constructive feedback. Due to limited time we are sorry to not reply to your questions. We have concluded that a different venue would be better suited for this work with additional adjustments. Thank you again for your time and comments these are deeply appreciated.

---

### Note · Authors · 2024-12-06

I have read and agree with the venue's withdrawal policy on behalf of myself and my co-authors.